# Deep Generative Models for Geometric Design Under Uncertainty

## Wei (Wayne) Chen, Doksoo Lee, Wei Chen

Department of Mechanical Engineering
Northwestern University
Evanston, IL 60208
wei.wayne.chen@northwestern.edu, doksoolee2024@u.northwestern.edu, weichen@northwestern.edu

## Abstract

Deep generative models have demonstrated effectiveness in learning compact and expressive design representations that significantly improve geometric design optimization. However, these models do not consider the uncertainty introduced by manufacturing or fabrication. Past work that quantifies such uncertainty often makes simplified assumptions on geometric variations, while the "real-world" uncertainty and its impact on design performance are difficult to quantify due to the high dimensionality. To address this issue, we propose a Generative Adversarial Network-based Design under Uncertainty Framework (GAN-DUF), which contains a deep generative model that simultaneously learns a compact representation of nominal (ideal) designs and the conditional distribution of fabricated designs given any nominal design. We demonstrated the framework on two real-world engineering design examples and showed its capability of finding the solution that possesses better performances after fabrication.

## Introduction

Many engineering design problems boil down to geometric optimization. However, geometric optimization remains a grand challenge because of its extreme dimensional complexity and often hard-to-achieve performance objective. Recent work has shown that deep generative models can learn a compact and expressive design representation that remarkably improves geometric design optimization performances (indicated by both the quality of optimal solutions and the computational cost) (Chen, Chiu, and Fuge 2020; Chen and Ramamurthy 2021; Chen and Ahmed 2021). However, past work based on deep generative models only considers the ideal scenario where manufacturing or fabrication imperfections do not occur, which is unrealistic due to the existence of uncertainties in reality, such as limited tool precision or wear. Such imperfections sometimes have a high impact on a design's performance or properties. Consequently, the originally optimal solution might not possess high performance or desired properties after fabrication.

Past work has developed non-data-driven robust optimization techniques to identify geometric design solutions that are insensitive to variations of load, materials, and geometry (Chen, Chen, and Lee 2010; Chen and Chen 2011; Wang

et al. 2019). However, due to the lack of generalized uncertainty representation that is compatible with the geometric representations, previous works often make simplified assumptions on geometric variations (*e.g.*, the distribution or the upper/lower bound of uncertain parameters), while the "real-world" geometric uncertainty and its impact on design performance are difficult to quantify due to the high-dimensionality. In this paper, we propose a *Generative Adversarial Network-based Design under Uncertainty Framework (GAN-DUF)* to allow uncertainty quantification (UQ) of geometric variability under real-world scenarios. This framework is generalizable to both shape and topology designs, and improves existing geometric design under uncertainty from four aspects: 1) The generative adversarial network (GAN) uses a compact representation to reparameterize geometric designs, allowing accelerated optimization; 2) The GAN associates fabrication uncertainty with ideal designs (*nominal designs*) by learning a conditional distribution of fabricated designs given any nominal design; 3) The optimization process accounts for the real-world distribution of geometric variability underlying any manufacturing processes, and allows UQ for robust design optimization or reliability-based design optimization; and 4) The compact representation of nominal designs allows efficient gradient-free global optimization.

We list the contributions of this work as follows:

1. We propose a novel deep generative model to simultaneously learn a compact representation of designs and quantify their real-world geometric uncertainties.

2. We combine the proposed model with a robust design optimization framework and demonstrate its effectiveness on two realistic robust design examples.

3. We build two benchmark datasets, containing nominal and fabricated designs, which will facilitate future study on data-driven design under manufacturing uncertainty.

## Background

In this section, we introduce Generative Adversarial Networks and previous work on design under uncertainty.

### Generative Adversarial Networks

The generative adversarial network (Goodfellow et al. 2014) models a game between a *generator G* and a *discriminator D*. The goal of $G$ is to generate samples (designs in our case)

that resemble those from data; while $D$ tries to distinguish between real data and generated samples. Both models improve during training via the following minimax optimization:

$$\min_G \max_D V(D, G) = \mathbb{E}_{\mathbf{x} \sim P_{\text{data}}}[\log D(\mathbf{x})] +$$
$$\mathbb{E}_{\mathbf{z} \sim P_{\mathbf{z}}}[\log(1 - D(G(\mathbf{z})))], \quad (1)$$

where $P_{\text{data}}$ is the data distribution and $\mathbf{z} \sim P_{\mathbf{z}}$ is the noise that serves as $G$'s input. A trained generator thus can map from a predefined noise distribution to the distribution of designs. Due to the low dimensionality of $\mathbf{z}$, we can use it to control the geometric variation of high-dimensional designs in design optimization. However, standard GANs do not have a way of regularizing the noise; so it usually cannot reflect an intuitive design variation, which is unfavorable in many design applications. To compensate for this weakness, the InfoGAN encourages interpretable and disentangled latent representations by adding the *latent codes* $\mathbf{c}$ as $G$'s another input and maximizing the lower bound of the mutual information between $\mathbf{c}$ and $G(\mathbf{c}, \mathbf{z})$ (Chen et al. 2016). The mutual information lower bound $L_I$ is

$$L_I(G, Q) = \mathbb{E}_{\mathbf{c} \sim P(\mathbf{c}), \mathbf{x} \sim G(\mathbf{c}, \mathbf{z})}[\log Q(\mathbf{c}|\mathbf{x})] + H(\mathbf{c}), \quad (2)$$

where $H(\mathbf{c})$ is the entropy of the latent codes, and $Q$ is the auxiliary distribution for approximating $P(\mathbf{c}|\mathbf{x})$. The Info-GAN's training objective becomes:

$$\min_{G,Q} \max_D \mathbb{E}_{\mathbf{x} \sim P_{\text{data}}}[\log D(\mathbf{x})] +$$
$$\mathbb{E}_{\mathbf{c} \sim P_{\mathbf{c}}, \mathbf{z} \sim P_{\mathbf{z}}}[\log(1 - D(G(\mathbf{c}, \mathbf{z})))] - \lambda L_I(G, Q), \quad (3)$$

where $\lambda$ is a weight parameter. In practice, $H(\mathbf{c})$ is usually treated as a constant as $P_{\mathbf{c}}$ is fixed.

## Design under Uncertainty

Design under uncertainty aims to account for stochastic variations in engineering design (*e.g.*, material, geometry, and operating conditions) to identify optimal designs that are robust or reliable (Maute 2014). Two common approaches are robust design optimization (RDO) and reliability-based design optimization (RBDO). RDO approaches simultaneously maximize the deterministic performance (or minimize the cost) and minimize the sensitivity of the performance (or cost) over random variables. The problem is typically formulated as (Chen and Chen 2011):

$$\min_{\mathbf{x}} J(\xi, \mathbf{u}(\mathbf{x})) = \mu(C(\mathbf{x}, \mathbf{u}(\mathbf{x}))) + k\sigma(C(\mathbf{x}, \mathbf{u}(\mathbf{x}))), \quad (4)$$

where $\mathbf{x}$ is the design variable, $\xi$ is the random variable; $\mathbf{u}$ is the state variable involved with the physics of interest, $C$ is the deterministic cost function. The mean cost is $\mu(C(\mathbf{x}, \mathbf{u}(\mathbf{x})) = \int_\xi p(\xi)C(\mathbf{x}, \mathbf{u}(\mathbf{x}))d\xi$ and the variance is $\sigma(C(\mathbf{x}, \mathbf{u}(\mathbf{x})))^2 = \int_\xi p(\xi)[C(\mathbf{x}, \mathbf{u}(\mathbf{x}) - \mu(C(\mathbf{x}, \mathbf{u}(\mathbf{x}))]^2 d\xi$. $k$ is the tuning parameter that adjusts the trade-off between the mean and variance of the cost function.

RBDO methods exploit stochastic methods to perform design optimization for a specified level of risk and reliability. A typical formulation reads (Maute 2014):

$$\min_{\mathbf{x}} \Pr(C(\mathbf{x}, \mathbf{u}(\mathbf{x})) \geq C^*)$$
$$\text{s.t.: } \Pr(f_m < 0) \leq \alpha^* \quad (5)$$

where $C^*$ is a tolerable threshold, $f_m < 0$ denotes failure in the system of interest, and $\alpha^*$ is the maximum acceptable failure probability.

Both approaches have facilitated design optimization under geometric uncertainty for various levels of geometric complexity (*i.e.*, size, shape, and topology). Among them, design optimization with topology variation under geometric uncertainty has been regarded as highly challenging due to modeling of topological uncertainty, propagation thereof, stochastic design sensitivity analysis, and others (Chen and Chen 2011). Our proposed model can overcome this challenge by using a deep generative model to learn arbitrary typologies and uncertainty distributions. We will demonstrate this capability using a real-world design example.

## Methodology

Let $\mathcal{I}_{\text{nom}}$ and $\mathcal{I}_{\text{fab}}$ denotes the datasets of nominal and fabricated designs, respectively:

$$\mathcal{I}_{\text{nom}} = \left\{ \mathbf{x}_{\text{nom}}^{(1)}, ..., \mathbf{x}_{\text{nom}}^{(N)} \right\}$$
$$\mathcal{I}_{\text{fab}} = \left\{ \left( \mathbf{x}_{\text{fab}}^{(1,1)}, ..., \mathbf{x}_{\text{fab}}^{(1,M)} \right), ..., \left( \mathbf{x}_{\text{fab}}^{(N,1)}, ..., \mathbf{x}_{\text{fab}}^{(N,M)} \right) \right\},$$

where $\mathbf{x}_{\text{fab}}^{(i,j)}$ is the $j$-th realization (fabrication) of the $i$-th nominal design. The **goals** are to 1) learn a lower-dimensional, compact representation $\mathbf{c}$ of nominal designs to allow accelerated design optimization and 2) learn the conditional distribution $P(\mathbf{x}_{\text{fab}}|\mathbf{c})$ to allow the quantification of manufacturing uncertainty at any given nominal design (represented by $\mathbf{c}$).

To achieve these two goals, we propose a generative adversarial network shown in Fig. 1a. Its generator $G$ generates fabricated designs when feeding in the parent latent vector $\mathbf{c}_p$, the child latent vector $\mathbf{c}_c$, and noise $\mathbf{z}$; whereas it generates nominal designs simply by setting $\mathbf{c}_c = \mathbf{0}$. By doing this, we can control the generated nominal designs through $\mathbf{c}_p$ and the generated fabricated designs through $\mathbf{c}_c$. Given the pair of generated nominal and fabricated designs $G(\mathbf{c}_p, \mathbf{0}, \mathbf{z})$ and $G(\mathbf{c}_p, \mathbf{c}_c, \mathbf{z})$, the discriminator $D$ predicts whether the pair is generated or drawn from data (*i.e.*, $\mathcal{I}_{\text{nom}}$ and $\mathcal{I}_{\text{fab}}$). Similar to InfoGAN, we also predict the conditional distribution $Q(\mathbf{c}_p, \mathbf{c}_c|\mathbf{x}_{\text{nom}}, \mathbf{x}_{\text{fab}})$ to promote disentanglement of latent spaces and ensure the latent spaces capture major geometric variability (Chen, Chiu, and Fuge 2020). The GAN is trained using the following loss function:

$$\min_{G,Q} \max_D \mathbb{E}_{\mathbf{x}_{\text{nom}}, \mathbf{x}_{\text{fab}}}[\log D(\mathbf{x}_{\text{nom}}, \mathbf{x}_{\text{fab}})] +$$
$$\mathbb{E}_{\mathbf{c}_p, \mathbf{c}_c, \mathbf{z}}[\log(1 - D(G(\mathbf{c}_p, \mathbf{0}, \mathbf{z}), G(\mathbf{c}_p, \mathbf{c}_c, \mathbf{z})))] - \quad (6)$$
$$\lambda \mathbb{E}_{\mathbf{c}_p, \mathbf{c}_c, \mathbf{z}}[\log Q(\mathbf{c}_p, \mathbf{c}_c|G(\mathbf{c}_p, \mathbf{0}, \mathbf{z}), G(\mathbf{c}_p, \mathbf{c}_c, \mathbf{z}))].$$

As a result, $G$ decouples the variability of the nominal and the fabricated designs by using $\mathbf{c}_p$ to represent the nominal design (**Goal 1**) and $\mathbf{c}_c$ to represent the fabricated design of any nominal design. By fixing $\mathbf{c}_p$ and sampling from the prior distribution of $\mathbf{c}_c$, we can produce the conditional distribution $P(\mathbf{x}_{\text{fab}}|\mathbf{c}_p) = P(G(\mathbf{c}_p, \mathbf{c}_c, \mathbf{z})|\mathbf{c}_p)$ (**Goal 2**).

The trained generator allows us to sample fabricated designs given any nominal design, simply by sampling the

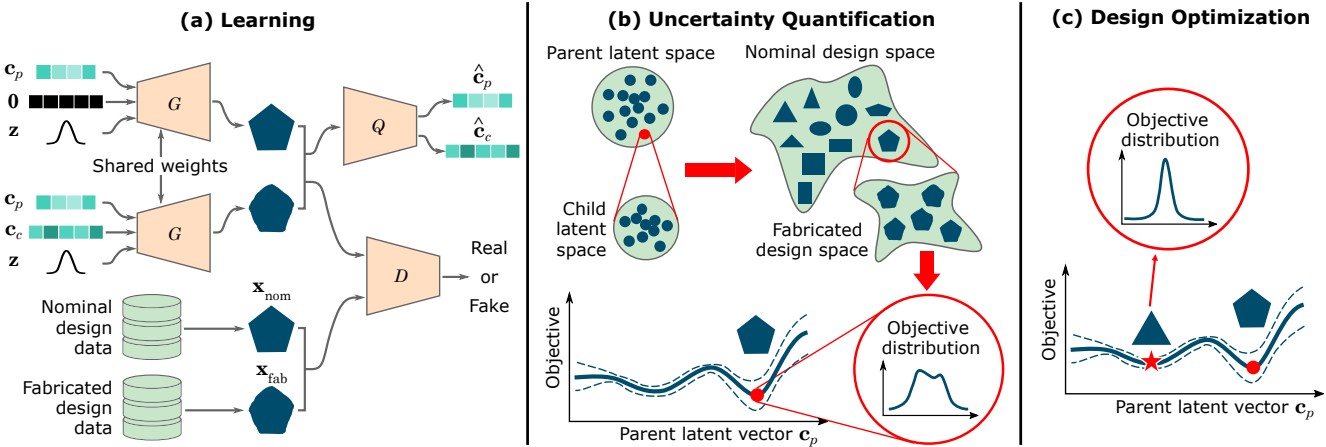

Figure 1: Illustration of proposed Generative Adversarial Network-based Design under Uncertainty Framework (GAN-DUF).

low-dimensional $\mathbf{c}_c$ with a fixed $\mathbf{c}_p$ representing the nominal design (Fig. 1b). We can then evaluate the objective(s) (*e.g.*, performance, quality, or properties) of these generated fabricated designs using computational methods (*i.e.*, physics simulation). The resulted distribution of objective(s) allows us to quantify the uncertainty for the nominal design. Note that the proposed framework is agnostic to both the type of designs (*e.g.*, how designs are represented or what geometric variability is presented) and downstream tasks like optimization. We can integrate the evaluated uncertainty into optimization frameworks including robust optimization, where we simultaneously optimize mean objective(s) and minimize the influence of uncertainty (Wang et al. 2019) (Fig. 1c), as well as reliability-based optimization, where we optimize the objective(s) subject to constraints such as failure probability or reliability index (Moustapha and Sudret 2019). The solution is expected to maintain high real-world performance or confidence of reliability even under fabrication imperfection.

## Experimental Results

We use the following two real-world robust design examples to demonstrate the effectiveness of our proposed framework.

### Airfoil Design

An airfoil is the cross-sectional shape of an airplane wing or a propeller/rotor/turbine blade. The shape of the airfoil determines the aerodynamic performances of a wing or a blade. We use the UIUC airfoil database[1] as our nominal design dataset $\mathcal{I}_{\text{nom}}$. Please refer to Appendix A for the pre-processing of $\mathcal{I}_{\text{nom}}$ and the creation of the fabricated design dataset $\mathcal{I}_{\text{fab}}$. The final dataset contains 1,528 nominal designs and 10 fabricated designs per nominal design. Note that due to the fact that similar nominal designs also have similar fabricated designs, we may need even fewer fabricated designs as training data. Studying the minimum required size of the fabricated design dataset might be an interesting future work.

We trained the proposed GAN on $\mathcal{I}_{\text{nom}}$ and $\mathcal{I}_{\text{fab}}$. Please refer to Appendix B for details on the model architecture

---

[1]http://m-selig.ae.illinois.edu/ads/coord_database.html

and training. We performed a parametric study to quantify the design space coverage and the uncertainty modeling performance of our trained models under different parent and child latent dimension settings. Details on the experimental settings and results are included in Appendix D. Based on the parametric study, we set the parent and the child latent dimensions of 7 and 5, respectively, when performing design optimization. The objective is to maximize the lift-to-drag ratio $C_L/C_D$ (please refer to Appendix C for details on design performance evaluation). We compared two scenarios:

1. Standard (nominal) optimization, where we only consider the deterministic performance of the nominal design. The objective is expressed as $\max_{\mathbf{c}_p} f(G(\mathbf{c}_p, \mathbf{0}, \mathbf{0}))$.
2. Robust design optimization, which accounts for the performance variation caused by manufacturing uncertainty. The objective is expressed as $\max_{\mathbf{c}_p} Q_\tau \left( f(G(\mathbf{c}_p, \mathbf{c}_c, \mathbf{0})) | \mathbf{c}_p \right)$, where $Q_\tau$ denotes the conditional $\tau$-quantile. We set $\tau = 0.05$ in this example.

In each scenario, we performed Bayesian optimization (BO) to find $\mathbf{c}_p$. We evaluate 21 initial samples of $\mathbf{c}_p$ selected by Latin hypercube sampling (LHS) (McKay, Beckman, and Conover 2000) and 119 sequentially selected samples based on BO's acquisition function of expected improvement (EI) (Jones, Schonlau, and Welch 1998). In standard optimization, we evaluate the nominal design performance $f(G(\mathbf{c}_p, \mathbf{0}, \mathbf{0}))$ at each sampled point. In robust design optimization, we estimate the quantile of fabricated design performances $f(G(\mathbf{c}_p, \mathbf{c}_c, \mathbf{0}))$ by Monte Carlo (MC) sampling using 100 randomly sampled $\mathbf{c}_c \sim P(\mathbf{c}_c)$ at each $\mathbf{c}_p$. Figure 2 shows the optimal solutions and the distributions of ground-truth fabricated design performances[2] of these solutions. By accounting for manufacturing uncertainty, the quantile values for performances after fabrication are improved for the robust optimal design $\mathbf{x}_{\text{robust}}^*$, compared to the standard optimal design $\mathbf{x}_{\text{std}}^*$, even though the nominal performance of $\mathbf{x}_{\text{robust}}^*$ is worse than $\mathbf{x}_{\text{std}}^*$. This result illustrates the possibility that the solution discovered by standard optimization can have

---

[2]"Ground-truth fabricated design" refers to designs created by the same means by which the designs from $\mathcal{I}_{\text{fab}}$ were created.

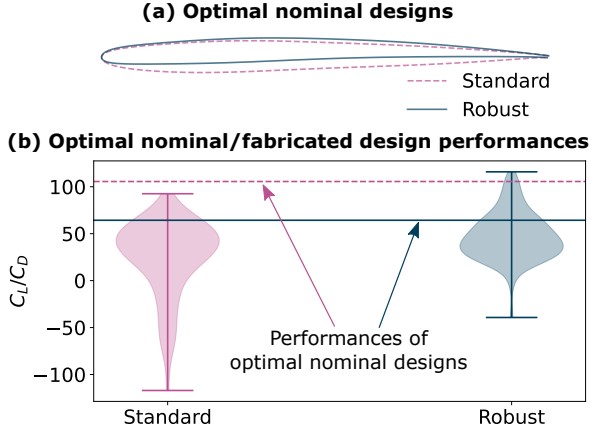

Figure 2: Solutions for the airfoil design example.

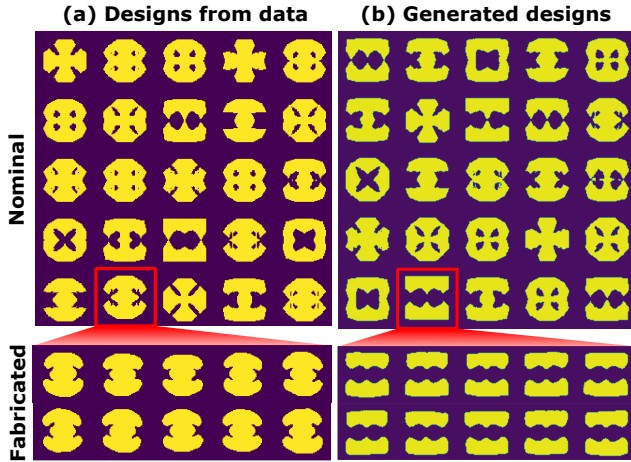

Figure 3: Metasurface designs randomly drawn from training data (a) and generated from a trained generator (b).

high nominal performance but is likely to possess low performance when it is fabricated. The robust design optimization enabled by GAN-DUF can avoid this risk.

## Optical Metasurface Absorber Design

Optical metasurfaces are artificially engineered structures that can support exotic light propagation using subwavelength inclusions (Chen, Taylor, and Yu 2016; Bukhari, Vardaxoglou, and Whittow 2019). Optical metasurface absorbers (Liu et al. 2017) have applications including medical imaging, sensing, and wireless communications. In this work, the key functionality of interest is large energy absorbance at a range of incident wave frequencies. Based on the method described in Appendix A, we created 1,000 nominal designs and 10 fabricated designs per nominal design (Fig. 3a).

As mentioned in the Background section, optimizing designs with varying topology under geometric uncertainty has been regarded as highly challenging (Chen and Chen 2011). GAN-DUF can handle this problem by modeling the uncertainty using the proposed generative adversarial network. Details on the model architectures and training can be found in Appendix B. Figure 3b shows nominal and fabricated designs randomly generated from the trained generator with a parent and a child latent dimensions of 5 and 10, respectively. We performed a similar parametric study, as in the airfoil design example, to quantify the design space coverage of the trained models under varying parent latent dimensions.

During the design optimization stage, we set the parent and the child latent dimensions to be 5 and 10, respectively. The objective is to maximize the overall absorbance over a range of frequencies (please refer to Appendix C for details). We compared standard optimization with robust design optimization. Due to the higher cost of evaluating the objective, we used fewer evaluations than in the airfoil design case. In each scenario, we performed BO with 15 initial LHS samples and 85 sequentially selected samples based on the acquisition strategy of EI. The quantile of fabricated design performances at each $c_p$ was estimated from 20 MC samples. Figure 4 shows the optimal solutions and the distributions of ground-truth fabricated design performances for these solutions. We observe similar patterns as in the airfoil design case,

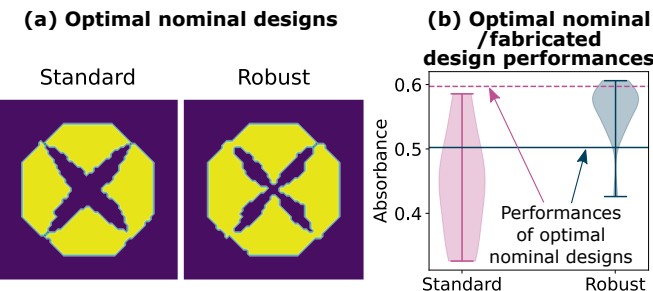

Figure 4: Solutions for the metasurface design example.

where the standard optimization finds the solution with higher nominal performance, while robust optimization enabled by GAN-DUF finds the solution with higher performances (in general) after fabrication. Note that the effect of robust design optimization is more significant on metasurface designs (Fig. 4b) than airfoil designs (Fig. 2b), which indicates a difference in the levels of variation in design performance sensitivity to manufacturing uncertainties. This difference can be caused by various factors such as the variance in nominal designs and the physics governing design performances.

## Conclusion

We proposed GAN-DUF to facilitate geometric design under manufacturing uncertainty. It contains a novel deep generative model that simultaneously learns a compact representation of nominal designs and the conditional distribution of fabricated designs given any nominal design. The proposed framework is generalizable as it does not make any assumption on the type of geometric representation or uncertainty. We applied GAN-DUF on two real-world engineering design examples and showed its capability in finding the design solution that is more likely to possess a better performance after fabrication. Built on these preliminary results, our future work will 1) perform more tests to quantify GAN-DUF's performance on different design under uncertainty scenarios and 2) use real fabricated designs as training and test data.

## Appendix A: Dataset Creation

In this appendix, we describe how we build the datasets of fabricated designs and nominal designs.

### Nominal Designs

**Airfoil Design.** The original UIUC database contains invalid airfoil shapes and the number of surface coordinates representing each airfoil is inconsistent. Therefore, we used the preprocessed data from Chen, Chiu, and Fuge (2020) so that outliers are removed and each airfoil is represented by 192 surface points (*i.e.*, $\mathbf{x}_{\text{nom}} \in \mathbb{R}^{192 \times 2}$).

**Optical Metasurface Absorber Design.** The nominal design dataset builds on three topological motifs — I-beam, cross, and square ring (Larouche et al. 2012; Azad et al. 2016). We create nominal designs by randomly interpolating the level-set fields of these baselines (Whiting et al. 2020). As a result, each design is stored as $64 \times 64$ level-set values (*i.e.*, $\mathbf{x}_{\text{nom}} \in \mathbb{R}^{64 \times 64}$). We can obtain final designs by thresholding the level-set fields. Building on a given set of baselines, this shape generation scheme allows a unit cell population that is topologically diverse.

### Fabricated Designs

Ideally, we can take the nominal designs from $\mathcal{I}_{\text{nom}}$, fabricate them, and use the fabricated designs as data. To save time and cost, we simulate the fabrication effects by deforming the geometry of nominal designs based on the following approaches.

**Airfoil Design.** We simulate the effect of manufacturing uncertainty by randomly perturbing the free-form deformation (FFD) control points of each airfoil design from $\mathcal{I}_{\text{nom}}$ (Sederberg and Parry 1986). Specifically, the original FFD control points fall on a $3 \times 8$ grid and are computed as follows:

$$\mathbf{P}_{\text{nom}}^{l,m} = \left( x_{\text{nom}}^{\min} + \frac{l}{7}(x_{\text{nom}}^{\max} - x_{\text{nom}}^{\min}), y_{\text{nom}}^{\min} + \frac{m}{2}(y_{\text{nom}}^{\max} - y_{\text{nom}}^{\min}) \right),$$
$$l = 0, ..., 7 \text{ and } m = 0, ..., 2, \tag{7}$$

where $x_{\text{nom}}^{\min}$, $x_{\text{nom}}^{\max}$, $y_{\text{nom}}^{\min}$, and $y_{\text{nom}}^{\max}$ define the 2D minimum bounding box of the design $\mathbf{x}_{\text{nom}}$. To create fabricated designs, we add Gaussian noise $\epsilon \sim \mathcal{N}(0, 0.02)$ to the $y$-coordinates of control points except those at the left and the right ends. This results in a set of deformed control points $\{\mathbf{P}_{\text{fab}}^{l,m} | l = 0, ..., 7; m = 0, ..., 2\}$. The airfoil shape also deforms with the new control points and is considered as a fabricated design. The surface points of fabricated airfoils are expressed as

$$\mathbf{x}_{\text{fab}}(u, v) = \sum_{l=0}^{7} \sum_{m=0}^{2} B_l^7(u) B_m^2(v) \mathbf{P}_{\text{fab}}^{l,m}, \tag{8}$$

where $0 \leq u \leq 1$ and $0 \leq v \leq 1$ are parametric coordinates, and the $n$-degree Bernstein polynomials $B_i^n(u) = \binom{n}{i} u^i (1 - u)^{n-i}$. We set the parametric coordinates based on the surface points of the nominal shape:

$$(\mathbf{u}, \mathbf{v}) = \left( \frac{\mathbf{x}_{\text{nom}} - x_{\text{nom}}^{\min}}{x_{\text{nom}}^{\max} - x_{\text{nom}}^{\min}}, \frac{\mathbf{y}_{\text{nom}} - y_{\text{nom}}^{\min}}{y_{\text{nom}}^{\max} - y_{\text{nom}}^{\min}} \right). \tag{9}$$

Perturbing nominal designs via FFD ensures that the deformed (fabricated) shapes are still continuous, which conforms to reality.

**Optical Metasurface Absorber Design.** Similar to the airfoil design example, we randomly perturb a set of $12 \times 12$ FFD control points in both $x$ and $y$ directions with white Gaussian noise that has a standard deviation of 1 pixel. This leads to the distortion of the $64 \times 64$ grid coordinates at all the pixels, together with the level-set value at each pixel. We then interpolate a new level-set field as the fabricated (distorted) design. To account for the limited precision of fabrication, we further apply a Gaussian filter with a standard deviation of 2 to smooth out sharp, non-manufacturable features.

Note that how well the simulated manufacturing uncertainty resembles the real-world uncertainty is not central to this proof of concept study. We treat the simulated uncertainty as the real uncertainty only to demonstrate our design under uncertainty framework. In the ideal scenario, we can directly use the real-world fabricated designs to build $\mathcal{I}_{\text{fab}}$ and our proposed framework can still model the fabricated design distribution give sufficient data, since the framework is agnostic to the form of uncertainty. However, one needs to use sufficient amount of data and appropriate dimensions for the latent vectors. For example, more fabricated design data and a higher-dimensional child latent vector are possibly required if the fabricated designs have a higher variation.

## Appendix B: Model Architectures and Training

In this appendix, we describe the model architectures and training configurations used in both examples.

**Airfoil Design.** We set the parent latent vector to have a uniform prior distribution $\mathcal{U}(\mathbf{0}, \mathbf{1})$ (so that we can search in a bounded space during the design optimization stage), whereas both the child latent vector and the noise have normal prior distributions $\mathcal{N}(\mathbf{0}, 0.5\mathbf{I})$. We fixed the noise dimension to 10, and experimented using different parent/child latent dimensions (please see Appendix D for the parametric study). The generator/discriminator architecture and the training configurations were set according to Chen, Chiu, and Fuge (2020). During training, we set both the generator's and the discriminator's learning rate to 0.0001. We trained the model for 20,000 steps with a batch size of 32.

**Optical Metasurface Absorber Design.** Same as the airfoil example, we set the parent latent vector to have a uniform prior distribution, while both the child latent vector and the noise have normal prior distributions. Again, we fixed the noise dimension to 10. The generator and the discriminator architectures are shown in Fig. 5. The discriminator predicts both the discriminative distribution $D(\mathbf{x}_{\text{nom}}, \mathbf{x}_{\text{fab}})$ and the auxiliary distribution $Q(\mathbf{c}_p, \mathbf{c}_c | \mathbf{x}_{\text{nom}}, \mathbf{x}_{\text{fab}})$. During training, we set both the generator's and the discriminator's learning rate to 0.0001. We trained the model for 50,000 steps with a batch size of 32.

## Appendix C: Design Performance Evaluation

During design optimization, the design performance is treated as the objective and needs to be evaluated at each iteration.

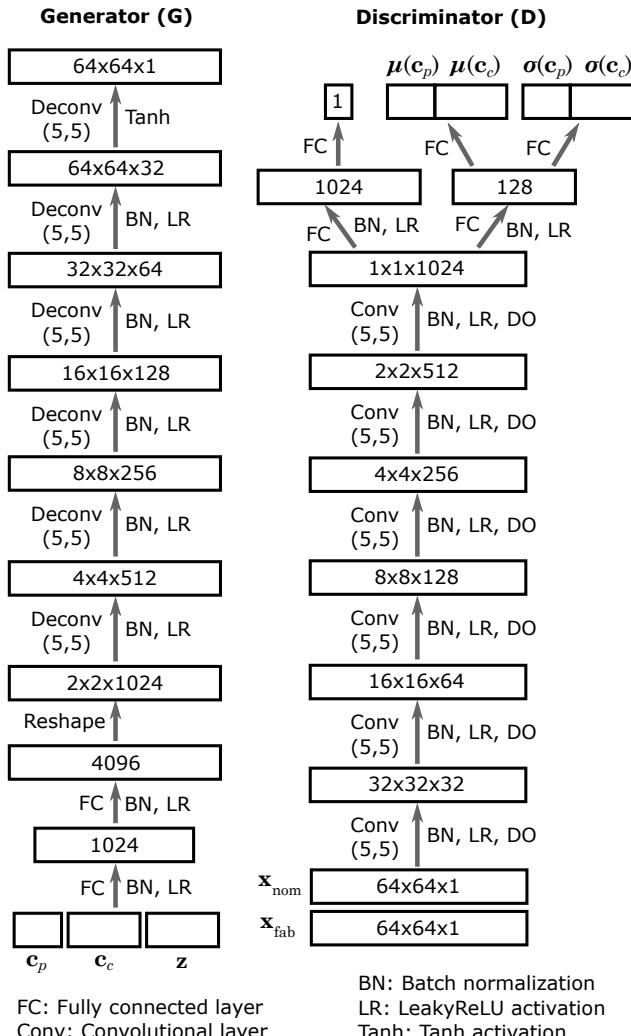

**Generator (G)**

64x64x1

Deconv (5,5) | Tanh

64x64x32

Deconv (5,5) | BN, LR

32x32x64

Deconv (5,5) | BN, LR

16x16x128

Deconv (5,5) | BN, LR

8x8x256

Deconv (5,5) | BN, LR

4x4x512

Deconv (5,5) | BN, LR

2x2x1024

Reshape

4096

FC | BN, LR

1024

FC | BN, LR

$\mathbf{c}_p$  $\mathbf{c}_c$  $\mathbf{z}$

**Discriminator (D)**

$\boldsymbol{\mu}(\mathbf{c}_p)$  $\boldsymbol{\mu}(\mathbf{c}_c)$  $\boldsymbol{\sigma}(\mathbf{c}_p)$  $\boldsymbol{\sigma}(\mathbf{c}_c)$

1

FC | FC | FC

1024 | 128

FC | BN, LR | FC | BN, LR

1x1x1024

Conv (5,5) | BN, LR, DO

2x2x512

Conv (5,5) | BN, LR, DO

4x4x256

Conv (5,5) | BN, LR, DO

8x8x128

Conv (5,5) | BN, LR, DO

16x16x64

Conv (5,5) | BN, LR, DO

32x32x32

Conv (5,5) | BN, LR, DO

$\mathbf{x}_{\text{nom}}$  64x64x1

$\mathbf{x}_{\text{fab}}$  64x64x1

FC: Fully connected layer
Conv: Convolutional layer
Deconv: Deconvolutional layer

BN: Batch normalization
LR: LeakyReLU activation
Tanh: Tanh activation
DO: Dropout

Figure 5: Generator and discriminator architectures in the metasurface design example.

In this appendix, we describe the details of the design performance evaluation for both examples.

**Airfoil Design.** An airfoil's aerodynamic performance is normally assessed by its lift and drag, which can be computed via a computational fluid dynamics (CFD) solver. In this paper, we used SU2 (Economon et al. 2016) as the CFD solver. The final performance is evaluated by the lift-to-drag ratio $C_L/C_D$.

**Optical Metasurface Absorber Design.** A unit cell of metasurface is made of a dielectric with relative permittivity $2.88\text{-}0.09i$ where $i$ is the imaginary unit $i = \sqrt{-1}$. Periodic boundary condition is imposed to the boundary of the analysis domain. The performance metric, energy absorbance, is defined as $A(f) = 1 - T(f) = 1 - |S_{11}(f)|^2$, where $f$ is the excitation frequency of an $x$-polarized incident wave (8-9 THz in this work), $T$ is the transmission, and $S_{11}$ is a

component of the $S$-parameter matrix that characterizes an electrical signal in a complex network. To achieve broadband functionality, we formulate the objective function as the sum of energy absorbance at individual frequencies (*i.e.*, $J = \sum_{i=1}^{n_f} A(f_i)$, where $n_f$ is the number of equidistant frequencies at which absorbance is to be observed).

## Appendix D: Parametric Study

We conducted parametric studies to investigate the effects of the parent and the child latent dimensions on the generative performances (we fix the noise dimension to 10). Particularly, we care about two performances: (1) how well the parent latent representation can cover nominal designs, and (2) how well the performance distributions of fabricated designs are approximated. The experimental settings and results are described as follows.

**Airfoil Design.** We evaluated the first performance (*i.e.*, nominal design coverage) via a fitting test, where we found the parent latent vector that minimizes the Euclidean distance between the generated nominal design and a target nominal design sampled from the dataset (*i.e.*, fitting error). We use SLSQP as the optimizer and set the number of random restarts to 3 times the parent latent dimension. We repeated this fitting test for 100 randomly sampled target designs under each parent latent dimension setting. A parent latent representation with good coverage of the nominal design data will result in low fitting errors for most target designs. Figure 6a indicates that a parent latent dimension of 7 achieves relatively large design coverage (low fitting errors). We evaluated the second performance (*i.e.*, fabricated design performance approximation) by measuring the Wasserstein distance between two conditional distributions — $P(f(\mathbf{x}_{\text{fab}})|\mathbf{x}_{\text{nom}})$ and $P(f(G(\mathbf{c}_p, \mathbf{c}_c, \mathbf{z}))|\mathbf{x}_{\text{nom}})$, where $f$ denotes the objective function. In this example, $f$ is the simulation that computes the lift-to-drag ratio $C_L/C_D$. For each generated nominal design $\mathbf{x}_{\text{nom}}$, we created 100 "simulated" fabricated designs as $\mathbf{x}_{\text{fab}}$, in the same way we create training data. We also generated the same number of fabricated designs using the trained generator. We compute the Wasserstein distance between these two sets of samples. We repeated this test for 30 randomly generated nominal designs under each child latent dimension setting. Figure 6b shows that when the child latent dimension is 5, we have relatively low Wasserstein distances with the smallest variation (the parent latent dimension was fixed to 7). When the child latent dimension further increases to 10, the uncertainty of the Wasserstein distances increase, possibly due to the higher dimensionality. Note that the training data only contains 10 fabricated designs per nominal design, while at the test phase we use many more samples per nominal design to faithfully approximate the conditional distributions. We do not need that many samples at the training phase because the generative model does not learn independent conditional distributions for each nominal design, but can extract information across all nominal designs.

**Optical Metasurface Absorber Design.** We performed a fitting test to study the effect of the parent latent dimension on the design space coverage of GANs. Same as in the airfoil

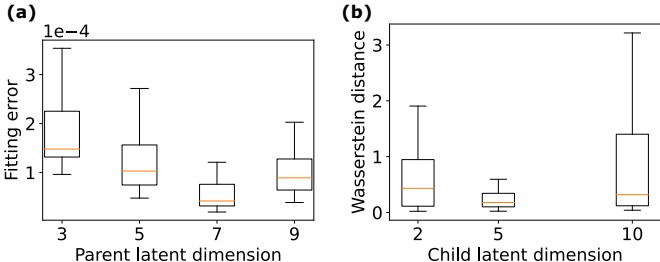

Figure 6: Parametric study for the airfoil design example.

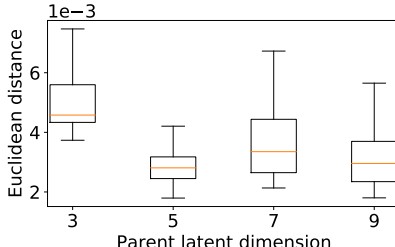

Figure 7: Parametric study for the metasurface design example.

design case, we use SLSQP as the optimizer and set the number of random restarts to 3 times the parent latent dimension. Here the fitting error is the Euclidean distance between the level-set fields of the generated nominal design and a target nominal design sampled from the dataset. Under each parent latent dimension setting, we randomly select 100 target designs. Figure 7 indicates that a parent latent dimension of 5 achieves sufficiently large design coverage, while further increasing the parent latent dimension cannot improve the coverage.

## Acknowledgement

This work was supported by the NSF CSSI program (Grant No. OAC 1835782). We thank the anonymous reviewers for their comments.

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
