# OpenReview forum: "Deep Generative Models for Geometric Design Under Uncertainty"
_AAAI.org/2022/Workshop/ADAM — AAAI 2022 Workshop ADAM_

### Official Review · Reviewer_PYQp · 2021-11-29
**A novel approach to Robust Design leveraging conditional generative models**

**Rating:** 9
**Confidence:** 5

**Review:**

This paper addresses the use of generative models for design under uncertainty, such as in the case where manufacturing variability may affect a design's performance or feasibility. The key idea of the paper is that rather than modeling design under uncertainty as a sequence of bounds or independent random variables, you can use a generative model to model the high dimensional covariance among the design parameters, and thus more accurately estimate the likely uncertainty. The other key idea in the paper, which I found quite compelling and novel, is to model this uncertainty as a conditional distribution over a given nominal design—that is, that one can directly learn how manufacturing (or other) variability is likely to arise given a target (i.e., nominal) design. This is a natural way of model uncertainty, since it goes directly from the "as designed" part to the "as made" part, and the properties of the generator can be usefully interrogated.

The paper itself uses a standard InfoGAN setup with a proposed weight sharing scheme for the "nominal" and "fabricated" shapes, and then uses the trained generators for Bayesian Optimization in both the standard and robust setting. It tests the model on airfoil and metasurface design examples, showing that, perhaps as expected, the robust designs possess higher performance than standard designs when subjected to manufacturing certainty. In addition, the paper provides a dataset of comparisons designs (nominal, fabricated) that can spur further developments along these lines for the community. The paper is well executed in the target scope for the workshop and has clear relevance to the workshop outcomes and goals; thus I think this is a good fit for this venue. One minor concern that the authors can consider as they move this work forward after the workshop is that, as written, the figures only compare the results in the standard and robust conditions for the *proposed GAN model* and not with respect to another compelling alternative (perhaps simpler) model or existing approaches to robust design. For this to be of wider archival use, you would need those comparisons, though for workshop discussion I think the scope and relevance of the workshop paper is fine as is.

---

### Official Review · Reviewer_tLTY · 2021-12-01
**The paper presents a Generative Adversarial Network-based design methodology which allows uncertainty quantification (UQ) of geometric variability. The main idea is to learn a low-D representation of possibly high-D nominal design spaces, and quantify the uncertainty through the learning of a conditional posterior distribution of the fabricated designs given any nominal design. The framework has been demonstrated on two design examples – airfoil design and optical metasurace absorber design.**

**Rating:** 9
**Confidence:** 4

**Review:**

•	The idea of using GANs to simultaneously learn the reduced design space and the conditional distribution quantifying the manufacturing uncertainty is novel.

•	Could you elaborate on the distinction between the variabilities present in the parent and the child latent space?

•	In some sense, the fabricated design space for a given nominal design involves heuristics in how the fabricated space is created (which is expected to be so – each problem is different). But it would be helpful to understand the effect of noise in creating the fabricated design space. For example, what happens when the standard deviation of the Gaussian noise is increased from 0.02 in the airfoil design fabricated space? What effect does it have in the optimization cycle in terms of evaluations/data requirement? If it is beyond the scope of this paper for a demonstration, it would be helpful to have some comments regarding this.

•	Any comments on how the initial samples were selected for the Bayesian optimization (BO) phase? As we know that BO results can be strongly influenced by the initial design. Also, what was the criterion for stopping the BO loop ?